# Purinergic Signaling in Oral Tissues

**DOI:** 10.3390/ijms23147790

**Published:** 2022-07-14

**Authors:** Mariachiara Zuccarini, Patricia Giuliani, Maurizio Ronci, Francesco Caciagli, Vanni Caruso, Renata Ciccarelli, Patrizia Di Iorio

**Affiliations:** 1Department of Medical, Oral and Biotechnological Sciences, University of Chieti-Pescara, Via dei Vestini 29, 66100 Chieti, Italy; mariachiara.zuccarini@unich.it (M.Z.); patricia.giuliani@unich.it (P.G.); patrizia.diiorio@unich.it (P.D.I.); 2Center for Advanced Studies and Technologies (CAST), University of Chieti-Pescara, Via L. Polacchi, 66100 Chieti, Italy; maurizio.ronci@unich.it (M.R.); f.caciagli@unich.it (F.C.); 3Department of Pharmacy, University of Chieti-Pescara, Via dei Vestini 29, 66100 Chieti, Italy; 4School of Pharmacy and Pharmacology, University of Tasmania, Hobart, TAS 7005, Australia; vanni.caruso@utas.edu.au; 5Stem TeCh Group, Via L. Polacchi, 66100 Chieti, Italy

**Keywords:** purines, purine receptors, purine enzymes, salivary glands, taste buds, dental structures

## Abstract

The role of the purinergic signal has been extensively investigated in many tissues and related organs, including the central and peripheral nervous systems as well as the gastrointestinal, cardiovascular, respiratory, renal, and immune systems. Less attention has been paid to the influence of purines in the oral cavity, which is the first part of the digestive apparatus and also acts as the body’s first antimicrobial barrier. In this review, evidence is provided of the presence and possible physiological role of the purinergic system in the different structures forming the oral cavity including teeth, tongue, hard palate, and soft palate with their annexes such as taste buds, salivary glands, and nervous fibers innervating the oral structures. We also report findings on the involvement of the purinergic signal in pathological conditions affecting the oral apparatus such as Sjögren’s syndrome or following irradiation for the treatment of head and neck cancer, and the use of experimental drugs interfering with the purine system to improve bone healing after damage. Further investigations are required to translate the results obtained so far into the clinical setting in order to pave the way for a wider application of purine-based treatments in oral diseases.

## 1. Introduction

Purines are ancestral molecules present in virtually all cell types. The discovery of their crucial function in cell biology dates back to 1953, when Watson and Crick published an article with the structure of DNA [1], a molecule that had been isolated in 1869 by Friedrich Miescher [2]. From the second half of the 19th century onwards, a huge number of studies has revealed the role of these compounds as fundamental intracellular constituents involved in numerous biochemical reactions and biological functions [3]. However, in the early 1970s, the pioneering studies carried out by Geoffrey Burnstock [4] pointed out the activity of these compounds as signal molecules. Since then, the influence of purines at extracellular level has been extensively elucidated on a wide variety of physiological and pathological processes in major organ systems including the central and peripheral nervous systems, gastrointestinal, cardiovascular, respiratory, renal, and immune systems [5,6,7,8,9,10,11,12]. However, the role of the purine system in the oral cavity is still not fully elucidated.

In humans, the oral cavity is bounded by teeth, tongue, hard palate, and soft palate [13]. A revision of the literature on the involvement of the purinergic system in the functioning/alterations of the tissues/cells which constitute this apparatus dates back to 2012 [14]. Since then, few other reviews have addressed the activity of purines on specific functions of the oral cavity [15,16] without a unifying framework. Thus, the purpose of this overview is to provide insight into the pathophysiological role of the purinergic signal in the oral cavity, pointing out new potential therapeutic targets in oral diseases.

## 2. Brief Outline of the Purine System as Intercellular Signal Network

In order to fulfill the large variety of the aforementioned effects [17,18], purines, in particular adenine-based compounds, must be present at extracellular level where they act as signal molecules.

ATP is constitutively released from all cells in physiological conditions and upon different types of stimulation [19], which remarkably increase the extracellular levels of the nucleotide from the basal ones (concentrations in the low nanomolar range) [19,20,21]. The mechanisms through which adenine-based nucleotides are released are numerous, including exocytosis, mainly from nerve terminals and chromaffin cells but also from different types of non-neuronal cells [22,23,24,25], and efflux through connexin/pannexin hemichannels and facilitated diffusion by nucleotide-specific ATP-binding cassette (ABC) transporters and multiple organic anion transporters (Figure 1) [26].

Adenosine, mostly derived from ATP metabolism, is also found in extracellular fluid. Occasionally, it can be released under physiological conditions, as it has been reported for neuronal cells, kidney cells, cardiomyocytes and immune cells [29]. However, adenosine release mainly occurs under cell stress conditions via nucleoside transporters, which are ubiquitously expressed in cell membranes but are generally deputed to recover extracellular adenosine to replenish the intracellular purine pool. These molecules are distinguished as equilibrative (ENT) and concentrative (CNT) transporters [27] (Figure 1). In contrast, nucleoside analogues are substrates for other carrier proteins which are mostly deputed to function as efflux transporters [28].

Finally, while there may be a transient ATP efflux under various stimuli perturbing cell membranes without evident damage, which include shear stress, hypotonic swelling, stretching, hydrostatic pressure, brief hypoxia/hypoglycemia/ischemia episodes as well as in response to Ca^2+^ agonists [30,31], all endogenous nucleotides and nucleosides can be discharged into the extracellular space following persistent membrane injuries. Thus, a massive leakage of these molecules occurs upon cell lysis due to organ injury, traumatic shock or inflammatory conditions [31], but also in the case of endothelial dysfunction with plaque formation, or as a consequence of bacterial and viral infections or intoxication by bacterial toxins, allergen contact or mechanical stimulation [22,29,32].

Purine release in the extracellular space represents a critical step for the initiation of purinergic signaling events. Indeed, nucleotides and adenosine activate different types of specific receptors, P1 receptors (primarily activated by adenosine) and P2 receptors (responding to both adenine and/or uracil nucleotides), which are widely but differentially expressed in virtually all cell types [26,32,33]. For more details, see Figure 2.

Noteworthy, the concentration of extracellular purines and pyrimidines is under the tight control of a multitude of metabolizing enzymes, which mainly function as membrane-bound enzymes (ectoenzymes) [38] and are listed as for names and functions in the Table 1. Altogether, these enzymes modulate the nucleotide signal termination, balancing purines concentration in the extracellular environment.

Thus, all players in this complex network contribute to a fine-tuning regulation of the purinergic signals in physiological conditions. However, these control mechanisms can be altered in several pathological conditions [3,39,40].

**Table 1 ijms-23-07790-t001:** Cell membrane-bound purinergic enzymes degrading extracellular adenine-based purines.

Family	Name	Function	References
Nucleoside triphosphate diphosphohydrolases (NTPDases)	NTPDase 1/CD39	ATP → AMP	[41]
	NTPDase 2	ATP → ADP (sustained accumulation)	
	NTPDases 3 and 8	ATP → ADP (transient accumulation)	
Ectonucleotide pyrophosphatases/phospho- diesterases (ENPPs)	ENPP1	ATP → AMP and PPi	[42]
	ENPP3	ATP → ADP	
Alkaline phosphatases (APs)	Tissue-specific AP (TNAP)	PPi → Pi (mainly in mineralized tissues)	[43]
		AMP → ADO	[44]
Acid phosphatases	Prostatic acid phosphatase (PAP)	AMP → ADO	[45]
Ecto-5′-nucleotidase	e-5′-NT/CD73	AMP → ADO	[38]
Ecto- adenosine deaminase	e-ADA	ADO → INO	
Purine-nucleoside phosphorylase	e-PNP	INO → HYPO	

The principal family of ATP metabolizing enzymes is represented by the nucleoside triphosphate diphosphohydrolases (NTPDases) including eight members. Of these, only NTPDase1, 2, 3, and 8 are cell surface-bound enzymes, with different activities. Extracellular ATP can also be metabolized by enzymes belonging to other families of enzymes such as ENPP and APs. AMP, derived from ATP metabolism, can be converted into adenosine by a member of the acid phosphatase superfamily, known as PAP, or by ecto-5′-nucleotidases (e-5′-NT, also known as CD73). Once formed, adenosine is degraded to inosine and then to hypoxanthine by the combined activity of cell surface-located enzymes, i.e., ADA or PNP, or transported into the cell by specific transporters to replenish adenine nucleotide pool. Arrows indicate the conversion of a compound into another one.

## 3. Purinergic Signals in the Gustatory System

The sense of taste is due to specialized epithelial cells within taste buds which are present in the tongue, palate and larynx. These cells exhibit different morphological and physiological properties so that they can be distinguished into four groups (Type I–IV cells) with Type II and Type III functioning as taste receptor cells [46]. For a long time, there was a debate on the nature of the neurotransmitters released by these cells, but in 2005 Finger et al. [47] indicated ATP as the main neurotransmitter allowing the communications between taste buds and gustatory nerves. It was also demonstrated that taste cells express a vesicular nucleotide transporter (VNUT) [48] that is responsible for the vesicular storage of ATP [49]. Further studies have demonstrated that ATP is released from taste buds, in particular from Type II cells transducing bitter, sweet and umami perception, and that P2X2/P2X3 receptors are required for the transmission of all taste quality, including those mediated by Type III cells (sour). Moreover, an ecto-ATPase, namely NTPDase2, was identified in the taste buds [50] as deputed to nucleotide metabolism; this enzyme activity also avoids P2X2/3 receptor desensitization [51].

The discovery of the mechanism leading to ATP release from these cells has been even more complex. Since Type II taste cells do not express components of the SNARE (soluble N-ethylmaleimide-sensitive factor attachment protein receptor) complex, which normally assures a vesicular release of the adenine-based nucleotide, some researchers suggested an ion channel-mediated mechanism. Initially, it was hypothesized an involvement of pannexin 1 (Panx1), which afterwards was shown to be responsible for the ATP release from mitochondria, the latter being the main intracellular source of ATP released from taste bud cells reviewed in [15]. Next studies unequivocally demonstrated that taste-evoked ATP release occurs via calcium homeostasis modulator (CALHM) channels, which are voltage gated channels with large calcium conductance. While the subtype CALHM1 was primitively involved, more recent studies have demonstrated that another channel subtype, namely CALHM3, needs to be co-expressed together with CALHM1 to account for ATP release with appropriate electric characteristics from taste buds [52]. However, these mechanisms have been reported only in Type II taste cells [15].

Many other reports have shown the expression of adenosine receptors in the taste buds, wherein A_2B_ receptors are mainly expressed in sweet sensitive taste cells. This finding has been shown in rodents [53] and indirectly confirmed in humans, in whom caffeine, a known nonselective antagonist of adenosine receptors, mainly A_1_R and A_2A_R, selectively inhibited sweet taste perception in long-lasting manner. Indeed, the participants to this study received a solution containing caffeine on the first day and a quinine coffee solution on the second; interestingly, findings showed that caffeine suppressed both the sweetness of the coffee samples and the sucrose subsequently tested. However, the treatment had no effect on bitter, sour, salty, or umami perception [54]. Moreover, in patients with taste dysfunction, the oral treatment with theophylline, another methylxanthine like caffeine, improved this dysfunction [55]. The effect was due to the theophylline-promoted increase in salivary levels of ‘Sonic Hedgehog’ (Shh), which is a protein with important regulatory roles in taste system morphogenesis as well as in taste bud homeostasis and taste perception reviewed in [56]. This protein is present also in parotid glands from which it is released into saliva. Theophylline would act to increase transcription factors, e.g., cyclic AMP and cyclic GMP through the known and often invoked inhibition of cyclic nucleotide phosphodiesterases, enzymes deputed to cyclic nucleotide metabolism. However, there are several limitations in finding interpretation, including the small number of patients with taste dysfunction and the variability in the dysfunction etiology.

Thus, these data provide the proof that both caffeine and theophylline, can influence taste acting at different levels and also with different mechanisms. In this regard, the difference in the molecular structure between caffeine and theophylline, namely three methyl groups for caffeine and only two methyl groups for theophylline, could be accountable for the different mechanisms of action to explain their activity. Indeed, while both compounds modulate GABA receptor action and regulate intracellular calcium levels, besides the antagonism of adenosine receptors and phosphodiesterase inhibition [57], it was recently shown that caffeine has a role distinct from theophylline in gene regulation [58]. Therefore, methylxanthine may produce unpredictable effects on taste perception depending on the use of one or the other compound.

## 4. Purinergic Signaling in the Salivary Glands

Saliva is the liquid secreted by three pairs of salivary glands located in the oral cavity, which are the parotids with pure serous secretion, the submandibular and sublingual glands with mixed secretion, mainly serous the former and mucous the latter. Salivary glands comprise, in turn, acinar epithelial cells, which mainly secrete water and electrolytes in saliva, and ductal cells, able to control the fluid electrolyte concentrations while myoepithelial cells contribute to secrete saliva from acinar cells [59].

Saliva plays a crucial role in maintaining the health of the structures of the oral cavity, participates in the formation of the food bolus as an active part in the food digestion process and protects the oral cavity from possible infections [60]. In addition, saliva facilitates taste sensation and it is ever more used to determine markers of systemic diseases as well as to monitor drug intake both for therapeutic purposes and in case of some illicit/harmful drug assumption [61,62].

Salivary ATP levels can be increased by ATP released from cells and bacteria therein present and reduced by the activity of ectonucleotidases [63]. A recent review has revised literature about P2 receptor signaling in the salivary glands [16]. Some P2 receptors were initially discovered in rat submandibular glands (SMGs) and included both metabotropic (P2Y_1_, P2Y_2_) and ionotropic (P2X4, P2X7) receptor subtypes [64], while a major number of P2 receptors have been detected in acinar cells of mouse parotid glands such as P2X2,4,7 and P2Y_1,2,10,12,14_ [64]. ATP and adenosine receptors have also been described in human oral glands [14].

The role of ionotropic P2 receptors has been investigated in physiological and pathological conditions as for saliva production. P2X7R, which show a low affinity towards ATP and require high micromolar concentrations of nucleotide to be activated [65], are expressed in salivary ductal cells. Their stimulation regulates ionic currents and saliva volume by a mechanism dependent on the influx of extracellular calcium through the non-selective cation channel associated to the stimulation of P2X7R, which in turn rises [Ca^2+^] [66]. However, P2X7R activation can also modulate the saliva formation by inhibiting intracellular Ca^2+^ mobilization caused by muscarinic or substance P receptor agonists [67,68]. These data were obtained in rat submandibular acinar cells and for the inhibitory mechanism it was suggested an interference with binding of the autonomic agonists to their receptors [67]. In agreement with these data, cholinergic effect was significantly increased in parotid acinar cells from P2X7R-null mice [69]. Further experiments have corroborated these observations reviewed in [16].

P2X4R, which are activated by lower concentration of ATP than P2X7R, are also expressed in salivary acinar and ductal cells with a role yet to be fully defined. Some studies on P2X7R null mice have highlighted a weak saliva production attributable to P2X4R activity [70]. Subsequent investigation in salivary epithelium has also pointed out that P2X4R and P2X7R can form heteromeric channels while others have demonstrated a possible modulatory role of P2X4R on the P2X7R function, resulting in a reduction of P2X7R affinity to ATP, with a decreased production of saliva. However, these aspects are still being debated. Conversely, it seems clear that there is a synergism between P2X4R and beta-adrenoreceptors so that their activation enhanced the Ca^2+^ influx in mouse and human parotid acinar cells, thus increasing saliva production [71,72].

As for metabotropic P2 receptors, studies performed so far have shown that P2Y_1_R play a marginal role in salivary glands, being active mainly during gland development. Indeed, although they are expressed in adult rat SMGs, there is a decreased coupling by them to the usual G protein α_q11_ subunit, which leads to a lack of activity by these receptors during the adult life [73]. However, these data should be confirmed in mice and humans. Conversely, activation of P2Y_2_ receptors, initially identified in a cell line of human salivary gland cells and subsequently in other epithelia, seems to contribute to saliva secretion [74]. However, data are not univocal and further research is warranted to clarify the role of these receptors.

In contrast, the literature on the expression of adenosine receptors in the salivary glands and its activity on the saliva production is very limited. In 1986, the presence of adenosine A_1_ receptors was reported on rodent salivary glands [75]. Subsequently, it was shown that antagonism for this receptor stimulated mucin secretion caused by isoproterenol in rat submandibular acini [76]. Furthermore, Finkelberg and coll. [77] demonstrated that parotid gland A_1_ receptor activation induces a stimulatory effect on amylase release associated with increased production of cAMP and inositol phosphate accumulation via phospholipase C activation.

As shown for many other tissues and organs, extracellular nucleotides are metabolized to adenosine by ectonucleotidases. It was previously demonstrated that e-5′NT and NTPDase1, 2 and 3, but not NTPDase8, are expressed in rat SMG, mostly associated to microsomes [78]. Subsequently, this activity was predominantly attributed to substrate inactivation by rat and human NTPDase2 [79]. Furthermore, it has been reported that nanovesicles contained in rat saliva have the ability to hydrolyze ATP. The characterization of these extracellular vesicles (EVs) has shown the presence of NTPDase1, -2, and -3, together with e-5′NT [80]. Of note, since it was demonstrated that NTPDases are present in acinar and not in ductal cells [81,82], it was suggested that EV enzymes can be transported along ducts, contributing to the control of saliva production [83] and, more in general, to the homeostasis of the oral cavity.

## 5. Purinergic Signaling in Teeth and Periodontium

Teeth are very resistant mineralized structures of different shapes and sizes. They are located in the maxillary bones, inside cavities called alveoli (Figure 3).

All tooth structures have been investigated for the presence and function of the purinergic signal, even though some of them (mainly dental pulp and periodontal ligament) have gained a major interest.

Starting from the external structures forming periodontium such as gums, adenosine receptors A_1_, A_2A_, and A_2B_ (but not A_3_) are expressed in human gingival epithelial cells (GECs) and fibroblasts therein present. The involvement of adenosine in healthy condition of these cells has not been defined, while its role as anti-inflammatory agent has been circumstantiated, as detailed in the next chapter. However, it should be considered that the expression of adenosine receptors can be remarkably altered by inflammatory processes affecting the gum. Thus, in human gingival tissue with chronic periodontitis, the expression of A_1_R was decreased by 20%, that of A_2A_R was more than doubled and that of A_2B_R increased by about 4-fold compared to healthy gingiva [84]. This implies that the responses to ligands for these receptors may not be the same depending on the inflammatory condition of the tissue. The presence of P2 receptors has also been reported in GECs, mainly the expression of P2X 2, 4, 5, 6, and 7 receptors [85]. In this case too, the role of P2XR, mainly P2X4R and P2X7R subtypes, has been explored in relation to inflammatory conditions [86]. Unfortunately, the release of purines has not been evaluated from the cells mentioned above.

In contrast, for human fibroblasts obtained from the periodontal ligament an increased release of ATP was shown when they were submitted to the application of a mechanical stress, as for example under orthodontic procedures. The mechanisms involved in such a release have not been explored. However, ATP released in this way enhanced extracellular signal-regulated kinase 1/2 (ERK1/2) phosphorylation, likely acting on P2Y_4_R or P2Y_6_R, overexpressed in these cells together with P2Y_5_R [87]. Additionally, it was recently demonstrated that P2X7R are usually absent in human periodontal ligament and dental tissues, while they can be observed in odontoblasts, which are cells contained in the pulp of the tooth [88].

In these cells, the presence and activity of the purinergic signal have been further investigated. In rat cultured odontoblasts, ATP release was evaluated upon thermal stimulation and it was found that both pulp tissue and odontoblasts expressed VNUT, which was colocalized in intracellular vesicles together with membrane fusion proteins. Further experiments demonstrated that ATP released from odontoblasts by thermal stimulation can interact with P2X3R on axons in the pulp [89].

More recently, the expression of P2Y_2_R, P2Y_4_R and all P2XR subtypes was revealed by RT-PCR in dental pulp whole tissue, while in acutely isolated rat odontoblasts the expression of all receptors aforementioned was confirmed in a subset of odontoblasts, except that of some P2XR, namely P2X1,3,5R [90]. As expected, stimulation of these cells with 100 μM ATP caused an increase of [Ca^2+^]_i_. This effect was not abolished by thapsigargin-induced depletion of intracellular Ca^2+^ ions, which suggested ionotropic P2XR activity. Since the ATP effect was maintained in more than half of the cells tested, even in a bath solution without extracellular Ca^2+^, this finding confirmed the activity of functional G_q/11_-coupled metabotropic ATP receptors in a subset of odontoblasts [90].

The presence of purinergic receptors was observed also in human dental pulp cells (HDPCs) by Wang and coll. [91]. These cells express P2X3, P2X4, P2X5, P2X7, and all P2Y receptor subtypes. Interestingly, the stimulation of these cells with low ATP concentration (10 μM) enhanced HDPC proliferation, while higher doses (800 μM) arrested cell proliferation while inducing odontoblastic differentiation, which was coupled to ERK/MAPK activation, and increased expression of specific markers such as dentin matrix protein 1 (DMP1) and dentin sialophosphoprotein (DSPP). In addition, all adenosine receptors, except A_2A_R, are expressed by the same cells and concur to their odontoblastic differentiation [92].

The ability of HDPCs to differentiate into odontoblasts and also osteoblasts has been reported in a number of studies and indicates the abundance of stem/progenitor cells among these cells. The evidence of the presence of mesenchymal stromal cells (MSCs) in the dental pulp as well as in various dental structures has emerged in the last ten years. As with those deriving from other tissue/fluid sources, dental MSCs show self-renewal capacity and multi-differentiation potential and a growing attention is now paid to their characteristics for possible therapeutic applications [93]. Among the eight types of dental stem cells (DSCs) so far identified, MSCs from dental pulp (DPSCs), human exfoliated deciduous teeth (SHED), and periodontal ligament (PDLSCs) have more extensively been investigated as autologous stem cells to use for regenerative purposes and tissue engineering therapies reviewed by [94]. Accordingly, the investigation on the purinergic signal has also been performed in these cells. Iwamoto and colleagues [95] reported that ATP release from odontoblast precursors occurred via one of the three members of the pannexin family, pannexin 3 (Panx3), which was not expressed in mature odontoblasts. They also demonstrated that Panx3 ATP hemichannel contributed to the regulation of the odontoblast transition from proliferation to differentiation stimulating the AMP-activated protein kinase (AMPK) signaling pathway, at the same time inducing cell cycle arrest upregulating p21 expression. In contrast the expression of connexin 43 (Cx43) has been recently reported in ameloblasts and not in odontoblasts [96]. The expression and activity of this membrane protein are important for dental development so that Cx43 mutations, revealed in oculodentodigital dysplasia, cause, among others defects, dental abnormalities in humans. However, no connection has been reported about the involvement of Cx43 in the release of ATP and related effects, as shown for other cells [97].

As for the presence of purinergic receptors in DSCs, our group has found the expression of all four adenosine receptors in DPSCs. We also revealed that the stimulation of A_1_R increased their duplication and mainly their osteogenic differentiation activating the Wnt signal [98]. Another study, comparing the growth characteristics and colony forming efficiency of DPSCs and SHED, found an increased proliferative activity in SHED coupled to a major expression of CD73 and CD146 along the passage number in vitro. Importantly, CD73, in addition to being considered a mesenchymal marker, is also known as the e-5′NT. Thus, it could be conceivable that CD73 activity also as enzyme able to metabolize AMP into adenosine at extracellular level could prolong the proliferative activity of these cells [99]. As for P2 receptors, the expression of P2X4R, P2X6R, and P2X7R and also of P2Y_1_R and P2Y_11_ was found in DPSCs. Some of them participated in ATP-induced Ca^2+^-signaling. In fact, activation of P2Y_1_R and P2Y_11_R stimulated the phospholipase C/inositol triphosphate (PLC/IP_3_) pathway via G protein α_q11_ with an increased release of Ca^2+^ form endoplasmic reticulum. This effect was coupled to Ca^2+^ entry via the stromal interaction molecule 1 (STIM1)-dependent opening of ORAI1 channels and P2X7R-opened channels. Altogether, this complex mechanism led to enhanced cell migration promoted by extracellular ATP [100]. Later on, Zhang et al. [101] found that the same cells expressed also P2X3R as well as P2Y_2_R and P2Y_6_R and that all of them likely contribute with the other P2XR and P2YR to ATP-induced inward currents. The authors also suggested that the activity of P2YR was related to the survival and proliferation of DPSCs, while the inhibition of P2XR affected only their proliferation. Furthermore, the activity of ATP in these cells was modulated by the NTPDase3, the blockade of which enhanced both ATP-induced inward ion currents and DPSC survival/proliferation. Again, a Ca^2+^-permeable Piezo1 channel, recently identified as mechano-sensing ion channel also in dental pulp cells [102], was revealed in human DPSCs and its activation was held responsible for cell migration upon ATP release and activation of P2 receptors (not better identified) coupled to stimulation of the proline-rich tyrosine kinase 2 (PYK2) and mitogen-activated protein kinase (MEK)/ERK pathways [103].

Finally, the oral cavity is widely innervated. In physiological condition, the trigeminal nerve provides important information against irritant and painful stimuli via specialized receptors, which may undergo altered expression and activity under pathological conditions leading to painful syndromes [104]. Fibers from the trigeminal nerve innervate also dental pulp together with sympathetic and parasympathetic neurons, which altogether contribute to the induction and maintenance of the tooth pain [105]. In this network it has to be considered that purinergic receptors are expressed on nervous fibers innervating dental pulp and gingival structures. Their role is to cooperate with the principal sensitive neurons to signal noxious stimuli. Their expression and role are both illustrated in the next section.

## 6. Purinergic Signal in Pathological Conditions of the Oral Cavity

Cell stress or apoptosis due to traumatic, ischemic or infective events in the oral cavity can trigger the release of ATP, ADP and other nucleotides promoting inflammatory responses via P2 receptors. In particular, high levels of ATP are released from cells during inflammation and they mainly activate P2X7R inducing reactive oxygen species production as well as the release of IL-1β and other cytokines [106]. Subsequent activation of IL-1 receptors by IL-1β in surrounding cells induces P2Y_2_R upregulation and further downstream responses to ATP and UTP. These nucleotides, being the main agonists of P2Y_2_R, are able to promote the release of further inflammatory agents (MCP-1, also known as CCL2) from macrophages locally present (i.e., in the dental alveoli thus amplifying the inflammatory environment). In this way, the release of ATP or UTP in response to cellular stress can locally modulate a wide range of signaling pathways to fine-tune the tissue response to inflammatory stimuli. Of course, purine metabolizing enzymes, present at the extracellular level and also in salivary EVs [80], likely lead to a reduction in signaling of P2R, favoring increase of anti-inflammatory events through activation of P1R [14].

However, the most frequent cause of inflammatory changes in the oral cavity are infections caused by bacteria, viruses and even unicellular eukaryotes such as yeasts. Bacterial infections mainly involve the periodontal tissues, therein inducing a host response mediated by immune cells, which are the principal responsible for ATP release. This ATP interacts with P2X7R, whose expression is increased, for example, by acute infection due to *Porphyromonas gingivalis*, one of the pathogens most frequently associated with periodontitis, and activates the NOD-like receptor pyrin domain-containing protein 3 (NLRP3) inflammasome through a canonical pathway. Consequently, the activity of caspase-1 is promoted, which in turn induces the maturation and secretion of interleukin-1β (IL-1β) and other active ILs [107,108,109]. Additionally, ATP, again interacting with P2X7R on local fibroblasts or osteoblasts, also activates the expression of receptor activator of nuclear factor k-B ligand (RANKL), inducing the differentiation and activity of osteoclasts to modify the alveolar bone crest morphology along with cytokines such as IL-1β and tumor necrosis factors α (TNFα), known to promote osteoclastic bone resorption. Therefore, there is a growing interest in the host-pathogenic microorganisms interaction [110], also in relation to the evidence that extracellular ATP from any source is a key modulator that initiates the pathway of alveolar bone resorption and bone loss in patients with periodontal disease [111]. Thus, possible control of extracellular ATP levels should reduce bone loss in periodontitis.

However, in periodontal infection sustained by other pathogens such as *Fusobacterium nucleatum*, P2X7R stimulation also activates the non-canonical inflammasome pathway with increased activity by caspase-11, in turn causing IL-1β secretion [112]. Just as *P. gingivalis* is also capable of releasing a nucleoside diphosphate kinase (NDK) homolog, which in turn metabolizes extracellular ATP while preserving GECs and underlying cells from ATP-P2X7R signaling [111], caspase-11 appears to be required not only for inducing cell death during *F. nucleatum* infection, but also to help limit bacterial load during *F. nucleatum* infection in bone marrow-derived macrophages and in mouse gingival tissue coinfected with *P. gingivalis* and *F. nucleatum*. These data would indicate that the both caspase-1 and caspase-11 pathways activated by P2X7R are involved in the immune response against infections [112].

In this context, although the expression of adenosine receptors can vary as a consequence of pathological conditions of one or more oral tissues [84], a role for this nucleoside should be considered. Indeed, stimulation of the adenosine receptors in GECs increased inducible nitric oxide synthase (iNOS) activity with enhanced production of stable nitric oxide (NO_2_ and NO_3_) metabolites, which have anti-inflammatory/antibacterial activity [113]. As well, in gingival fibroblasts, agonists for A_1_AR and A_2A_AR were shown to synergistically increase IL-1β-induced IL-6 and IL-8 production, which corroborates the involvement of adenosine signaling in the regulation of inflammatory responses in periodontal tissues [114]. Adenosine also inhibited the action of pro-inflammatory chemokines such as CXCL8, which plays a central role in the inflammatory response driven by periodontal infection. Interestingly, adenosine effect was mediated by an increased activity of the intracellular hemeoxygenase-1 and adenosine monophosphate-activated protein kinase (AMPK), both being key regulators of the host inflammatory responses [115]. Furthermore, studies on A_2A_R have demonstrated that their stimulation may be used to induce replication of *P. gingivalis* in GECs [116], which may have a protective effect, as above reported. Even though specific studies on other oral tissues are still lacking, it should be underlined that the signaling of adenosine through its receptors (i.e., A_2A_R and A_2B_R) has been proposed as the major immunomodulator stimulus during inflammatory conditions, especially those caused by chronic microbial infections such as those occurring in the periodontium [117].

Accordingly, the activity of purine metabolizing enzymes, which modulates the concentrations of extracellular purines in the oral cavity, must also be considered. ATP hydrolysis was revealed many years ago [118] in rat dental tissues by specific enzymes including alkaline phosphatase, ATPases, 5′-nucleotidase and pyrophosphatases. Further data have shown the activity of these enzymes in pathological conditions of the oral apparatus. For example, an increased secretion of ectonucleotides was demonstrated from saliva nanovesicles in rat SMGs stimulated with histamine, the first agent triggering inflammatory events [119]. Preliminary results from the same group would indicate that in EVs isolated from human saliva ATPase activity was higher in patients with periodontal disease [80]. Thus, enzymes in the saliva EVs could help in local control of nucleotides in sites where they were acutely increased due to injury or inflammation. Again, ATP has been shown to indirectly inhibit IL-1β-induced matrix metalloproteinases (MMPs) through the activity of NTPD1/CD39 corroborated by that of CD73 in primary human gingival fibroblasts (HGF) [120], both leading to an increased formation of adenosine. Another enzyme, the purine nucleoside phosphorylase (PNP), which catalyzes the reversible phosphorolysis of purine nucleosides, i.e., inosine and guanosine into hypoxanthine and guanine, respectively, has resulted to be upregulated in gingival crevicular fluid in periodontal disease [121]. Of note, the inhibition of the activity of this enzyme halted bone loss consequent to periodontal disease in an animal model [122]. Altogether, these results indicate that the increased amount of extracellular ATP released as a consequence of inflammatory environment is metabolized by a chain of extracellular enzymes, whose concerted activity is oriented to reducing potential negative effects of purine compounds and restoring a homeostatic condition.

Clearly, inflammation is tightly coupled to exacerbation of the pain perception. There is a large amount of evidence that an increase in extracellular ATP in oral tissues can exacerbate pain associated with inflammation. Several purinergic receptors are expressed in trigeminal ganglion (TG) nerves, including P2X2R and P2X3R. In particular, it was found that ATP, released from mechanically stimulated odontoblasts via pannexin-1 in response to the activation of transient receptor potential (TRP) channels (which are involved in the painful stimulus detection and transmission) [123], transmits a signal to P2X3R on TG neurons. Thus, odontoblasts behave as sensory receptor cells while ATP released from them acts as a neurotransmitter for dentinal pain [124]. Again, P2X3R expressed in TG appear to be involved in hyperalgesia of the temporomandibular joints and massetere muscle consequent to the placement of an occlusal interference in rats [125]. Kawaguchi et al. [126] observed the expression of other ATP/P2R, specifically the P2Y_12_R, in TG neurons, while Suguwara et al. [127] have found these receptors in TG satellite cells, which are cells surrounding TG neurons involved in the modulation of TG transmission. The stimulation of P2Y_12_R on satellite cells leads to an interaction with calcitonin gene-related peptide (CGRP) neurons, thus contributing to enhance the neuropathic pain in the tongue due to lingual nerve injury in rats. Conversely, adenosine has an opposite role, being analgesic, anti-inflammatory and tissue protector [128,129]. Together with purinergic receptor expression, purine enzymes have been detected in nervous fibers innervating dental tissues. While NTPDase3 is expressed in TG nociceptive neurons [130], NTPDase2 is present in the dental pulp odontoblast layer and it likely derives from the Schwann cells that encapsulate the nerve fibers projecting into this layer [131]. The expression of CD73 has also been detected in TG nociceptive neurons [132]. Altogether, these results suggest that ecto-nucleotidases also participate in the modulation of nociceptive signals. All of these aspects are summarized in Table 2.

## 7. Clinical Perspectives

In dentistry, bone repair represents one of the main issues, especially in the presence of large-size bone defects. Regenerative medicine has introduced the experimental use of MSCs, although there are still several limits to overcome linked to accessibility, storage and expansion of these cells. Furthermore, it is even more evident that MSCs contribute to heal injured tissues through the release of factors contained in EVs released from MSCs. In particular, studies on these EVs and their content can facilitate their use in diverse animal models of calvarial and alveolar bone regeneration [136]. Recently, it was demonstrated that EVs derived from PDLSCs can be embedded in a hydrogel scaffold (i.e., matrigel) and then transplanted into a rat model of calvarial defect. The results showed that this device stimulated host cell proliferation and bone repair in vivo and this effect, inhibited in the presence of a nonselective adenosine receptor antagonist, theophylline, could be attributed also to adenosine-mediated activation of AKT and ERK1/2 pathways [137].

Accordingly, other findings obtained by Verma et al. [138] showed that an osteogenic scaffold comprising adenosine/epigallocatechin gallate-N,O-carboxymethyl chitosan/collagen type I (AD/EGCG-g-NOCC@clgn I), exhibited osteo-inductive properties in a calvarial defect model reproduced in BALB/c mice. The ex-vivo findings clearly established the pro-osteogenic potential of adenosine and EGCG, which both stimulated MSCs toward osteoblast differentiation as evaluated by the increased expression of selective osteogenic markers such as alkaline phosphatase and osteocalcin and enhanced calcium deposits. Noteworthy, this type of 3D matrix held some extracellular matrix (ECM) properties, providing a favorable microenvironment and a structural support against mechanical stress. 3D matrix also acted as a reservoir for the sustained release of osteo-inductive molecules such as adenosine. Thus, two different experimental approaches have confirmed/highlighted an important osteo-inductive role for this nucleoside. Of course, further studies are needed in this direction to open the way to new strategies for bone repair.

Interestingly, the use of a regenerative therapy has been attempted also to restore the function of salivary glands after radiation injury. By now, this possibility has been ascribed to the activation of muscarinic receptors through a mechanism dependent on the presence of progenitor cells also in salivary glands [139]. Given the expression of purinergic receptors in this district, it should be of interest to evaluate whether their modulation can play a role also in the regeneration of acinar cells.

However, other experimental findings seem to converge towards a useful application of the purinergic receptor modulation in salivary gland dysfunction, which is usually observed in the Sjögren’s syndrome (SS) and as a consequence of radiotherapy (RT) used for treating head and neck cancer (HNC). In both pathological conditions, hyposalivation is coupled to impairment of muscarinic receptor signal and aquaporin channel activity, which are required for saliva formation and fluid secretion, respectively [140,141]. Noteworthy, an inflammatory environment with an increased ATP release and P2X7R expression are present in both cases and play a fundamental role in the onset and evolution of these conditions [142]. Thus, as for SS and RT applied to HNC treatment, the use of antagonists for these receptors has been evaluated. In particular, inhibition of P2X7R by the competitive antagonist A-438079 significantly reduced sialadenitis, while improving carbachol-induced saliva flow in a murine model of SS-like syndrome [143]. Furthermore, in salivary gland biopsies from SS patients, P2X7R antagonism reduced the expression of upregulated immuno-active molecules such as IL-1β, intercellular adhesion molecule (ICAM), vascular cell adhesion molecule (VCAM), E-selectin, CD80, and CD86 [143]. As well, in vivo administration of A-438079 in γ-irradiated mice induced a significant radioprotection to salivary glands and maintained saliva flow rates similar to non-irradiated mice for a prolonged period after irradiation. This effect was coupled to a reduction in the secretion of prostaglandin E_2_ (PGE_2_), which is considered one of the most inflammatory markers following gamma-radiation therapy [144]. Altogether, these findings highlight the importance of P2X7R for a potential application of their ligands in both diseases. However, besides P2X7R, we cannot neglect the existence and the role of P2Y_2_R, whose upregulation has been observed under stress/inflammatory condition. Indeed, upregulation of these receptors but not that of other uridine nucleotide receptors such as P2Y_4_R and P2Y_6_R is present in SMGs of NOD.B10 mouse model of primary SS as compared to normal C57BL/6 mice [145]. In agreement with these findings, Jasmer et al. [146] have recently shown that inhibition of P2Y_2_R using the selective antagonist AR-C118925 in another animal model of SS resolved sialadenitis, while improving compromised salivary flow. Likewise, P2Y_2_R deletion ameliorated sialadenitis induced by IL-14α, an autoimmune syndrome with characteristics similar to SS [147].

Finally, while the research for purine enzyme dysregulation seems to be still in its infancy, the investigation of the presence of receptor polymorphisms looks more realistic and useful. Indeed, recent data support a relationship between four P2X7R genetic variants showing a gain of function and the amplitude of the inflammatory response in patients with aggressive periodontitis [148]. This is consistent with a study reporting a single polymorphism at the minor allele of P2X7R A1405G, which was associated with a gain of receptor function and may represent a risk factor in a subset of SS patients who do not express human leukocyte antigen (HLA) risk alleles [149]. Further findings on the variability of *P2X7R* gene suggest that this factor could contribute to the etiopathogenesis of post-orthodontic external apical root resorption, as investigated in two different populations of young people [150,151].

## 8. Conclusions

There is convincing evidence that purinergic receptors are expressed and functioning in oral tissues, contributing with their signals to the homeostatic regulation of the mechanisms that ensure the correct functionality of the oral cavity. Moreover, data so far collected as for therapeutic applications of purinergic compounds in oral pathologies are encouraging, although further studies are needed regarding their efficacy and long-term safety profile. Therefore, a cooperative approach is desirable to bridge the gap between basic and clinical research in the dental field and to hopefully pave the way for new clinical applications of ligands of purine receptors and/or enzymes in the management of diseases of the oral cavity.

## Figures and Tables

**Figure 1 ijms-23-07790-f001:**
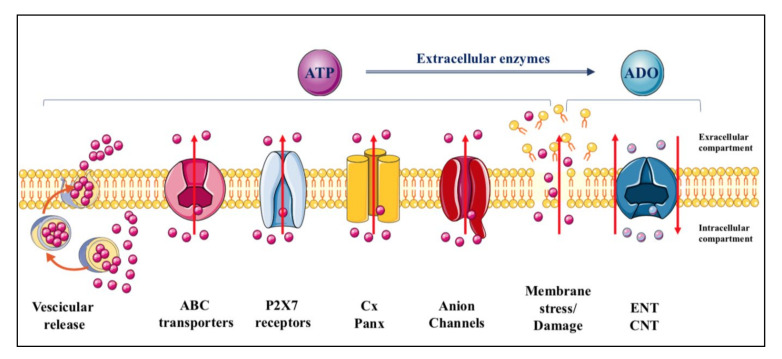
Mechanisms of purine release. ATP is released from cells in physiological conditions through multiple ways including vesicular exocytosis, connexin/pannexin (Cx/Panx) hemichannels, facilitated diffusion by nucleotide-specific ATP-binding cassette (ABC) transporters, and multiple organic anion transporters. In contrast, while most of the adenosine is formed from the nucleotide metabolism at extracellular level, the principal mechanism of adenosine efflux from cells is assured, under cell stress conditions, by carriers identified as equilibrative (ENT) and concentrative (CNT) nucleoside transporters. The former are bidirectional carriers, the direction of transport depending on the nucleoside concentration gradient across the plasma membrane, while the latter are Na^+^-dependent, being the nucleoside transport coupled to that of the sodium ion and independent of the nucleoside concentration gradient [27]. Additional transporters such as organic anion and cation transporters and ABC transporter proteins, have been implicated only as carriers of nucleoside-derived drugs, particularly those used as antiviral drugs [28].

**Figure 2 ijms-23-07790-f002:**
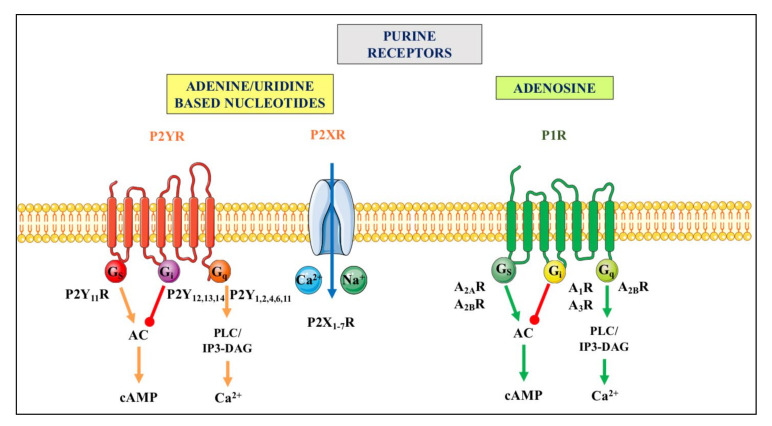
Purinergic receptors. They are divided in two principal families named P1 and P2. The P1 receptor (P1R) family includes four subtypes (A_1_R, A_2A_R, A_2B_R and A_3_R), which are all metabotropic G proteins-coupled receptors [34]. The P2 receptor (P2R) family is subdivided into seven ionotropic P2X (P2XR), which are activated by ATP, and eight metabotropic P2Y receptors (P2YR), of which P2Y_1_R respond to ATP and ADP, P2Y_2,4,6_R are mainly activated by uridine-based nucleotides, P2Y_12,13_R respond to ADP, and P2Y_14_R to UDP-glucose (as recently reviewed by [35,36]). Ionotropic P2XR, when stimulated, allow the entry of cations such as Na^+^ and or Ca^2+^ into cells while metabotropic receptors belonging to the P1R family or to the P2YR subtypes are coupled to different G proteins and downstream molecular pathways indicated above. For major details see [37]. Abbreviations: AC, adenylate cyclase; cAMP, cyclic AMP, PLC/IP3-DAG, phospholipase C/inositol triphosphate-diacylglycerol.

**Figure 3 ijms-23-07790-f003:**
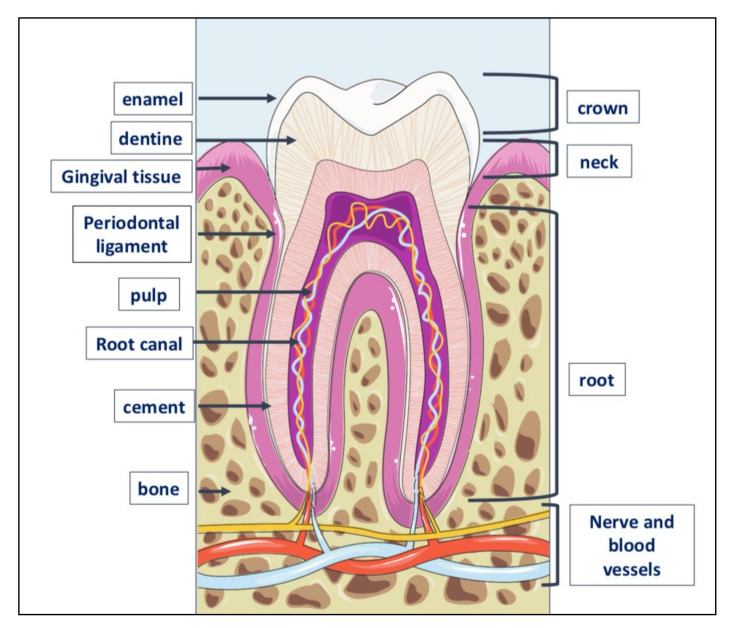
Tooth anatomy. The tooth consists of three parts: crown, the visible part of the tooth; neck or collar, the part surrounded by the gingiva or gum, and root located inside the alveolar bone to which is linked by the periodontal ligament. The outer part of the crown is covered with a layer of dental enamel, the hardest element in the human body; below the enamel is dentin, the substance that gives the teeth their color and composed of about 70% of inorganic material, the remaining 30% of organic material and water. Dentin is generated by odontoblasts, cells contained in the pulp, which is inside the dentin and includes also the nerve and blood vessels. The periodontium is a set of structures that are located around the tooth and which have a dual function: to keep the tooth firmly in the alveolar bone and to preserve the integrity of the mucous membrane of the oral cavity. It is made up of four tissues: the gum, the periodontal ligament, the root cement and the alveolar bone itself, that is, that bone, also called the hard lamina, which delimits the dental alveolus.

**Table 2 ijms-23-07790-t002:** Purinergic signaling in oral cavity pathologies.

Pathological Condition	Involvement of the Purinergic System	Experimental Model	Activated Mechanisms	References
Inflammation	P2X7R	Mouse SMG cell aggregates	Membrane blebbing, increase of caspase activity, cleavage and release of α-fodrin (a supposed auto-antigen important in SS development)	[106]
Inflammation in peridontal disease	P2X7R	In vitro and in vivo studies	Increased release of ATP by gingival fibroblasts and bacteria able to stimulate P2X7R. This event promotes leukocyte recruitment and the activation of a canonical pathway associated to NLRP3 inflammasome activation, followed by inflammatory cell secretion of cytokines, interleukin IL-1β, TNF and RANKL that further trigger alveolar bone resorption. These events have been observed mainly along periodontal infections due to *P. gingivalis*.	[107,108,109,110,111]
			Combined periodontal infection by *P. gengivalis* and *F. nucleatum* also activates P2X7R via ATP, additionally stimulating a non-canonical pathway coupled to the activity of caspase 11, which would also contribute to limit bacterial load.	[112]
Inflammation	A_1_R and A_2A_R	HGFs	Stimulation of the two receptors by agonists synergistically increased IL-1β-induced IL-6 and IL-8 production and inhibited the adherence of IL-1β-stimulated HGF to activated lymphocytes, thus regulating the inflammatory responses.	[114]
Inflammation	CD39, CD73 and adenosine receptors	HGFs	Stimulation of adenosine receptors with the nonselective agonist NECA inhibited, such as ATP, an IL-1β-induced CXCL8 secretion. This effect was mediated by activation of heme-oxygenase 1 (HO-1) and phosphorylated adeno-sine mono-phosphate–activated protein kinase (pAMPK). The inhibition of CD73 or adenosine receptors abrogated the ATP effect on CXCL8 secretion.	[115]
			The activity of CD39, metabolizing ATP and favoring adenosine format-ion, was implicated also in ATP-induced inhibition of MMP-1 expression in the same cells	[120]
Inflammation	Ectonucleotidases	EVs from rat and human saliva	Enzyme secretion due to histamine stimulation in rats.	[119]
			Enzyme secretion due to periodontal disease in humans.	[80]
Periodontal disease	PNP	Human gingival crevicular fluid	Presence of increased expression of the enzyme	[121]
		Rats	Enzyme inhibition in ligature-induced periodontal disease reduced bone loss	[122]
Painful sensation	P2X3R	Odontoblasts	Dentinal pain upon receptor stimulation	[124]
			Hyperalgesia in temporomandibular joints	[125]
Painful sensation	P2Y_12_R	Lingual nerve injury in rats	Involved in the neuropathic pain in the tongue	[126]
Painful sensation	NTPDases 3 and CD73	TG nociceptive neurons	All these enzymes are presumed to participate in the nociceptive modulation	[130,132]
	NTDPases 2	Dental pulp odontoblasts		[131]
Inflammation	P2Y_2_R	Duct-ligated rat SMG	Receptor upregulation caus-ed by increased ATP release that stimulated IL-1β and other cytokine release	[133]
Experimental infection coupled to inflammation	P2X7R/P2X4R cooperation	*P. gingivalis* infection of hGECs	Activation of the two receptors by ATP stimulated ROS production, NLRP3 inflammasome activation and IL-1β release	[134]
Periodontal surgery	P2X4R	Rat	Increased receptor express-ion consequent to surgical detachment of marginal gingiva. P2X4R stimulation by ATP released by damaged cells led to direct activation of osteoclasts on bone surface increasing bone loss	[135]

CD73, also known as e-5′NT, ecto-5′-nucleotidase; *F. nucleatum*, *Fusobacterium nucleatum*; HGFs, human gingival fibroblasts; MMP-1, matrix metalloproteinase-1; NLRP3, NOD-like receptor pyrin domain-containing protein 3; NTDPase, ecto-nucleotide triphosphate diphosphohydrolase; *P. gingivalis*, *Phorphyromonas gingivalis*; PNP, purine nucleoside phosphorylase; RANKL, receptor activator of nuclear factor kappa-B ligand; SMG, submandibular gland; SS, Sjögren’s syndrome; hGECs, human gingival epithelial cells; TG, trigeminal ganglion.

## Data Availability

Not applicable.

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
