# Peer review of "Purinergic Signaling in Oral Tissues"

_ijms, 2022, doi:10.3390/ijms23147790_

Round 1

Reviewer 1 Report

This is an intensive critical review about purinergic signaling in oral cavity.

It provide insight into the role and mechanism  of this signing in the process of biology, physiology and pathology changes  in oral cavity and dental tissues. It also highlights the potential therapeutic effects of this signaling in oral healthy and  dentistry.

The authors has  reviewed and  cited  most of the current  studies of puringic signaling in oral cavity.  Even though it doesn't cover all the aspects of purinegic signaling in oral cavity,  for example its role in the oral cavity development, odontogenesis, and effect and mechanism of ATP signing in tooth   movement and bone modification in the process of orthodontic treatment,  this review is systematic, concise,  and very well organized.          

Author Response

We thank this reviewer very much for his/her appreciation of our work. We are aware that we have not covered all the topics concerning the oral cavity, but it would have been a very difficult task with the risk of making the review itself too heavy. However, the indicated topics that we have left out might be a good suggestion for another paper.

We resubmit our article after correcting of the errors indicated. 

Reviewer 2 Report

The manuscript by Zuccarini et al., gives a detailed overview of the work on purinergic signalling in oral tissues.

The manuscript gives describes in-depth the role of purinergic signalling during various oral pathological conditions. Moreover, the authors discuss impressively the clinical perspectives of purinergic signalling.

The manuscript is well-written and properly designed.

Author Response

We thank this reviewer very much for appreciating our review. We revised it by correcting the spelling mistakes of the words and trying to make the whole text clearer.

Reviewer 3 Report

The authors propose update the literature regarding the presence and possible role of the purinergic system in the different structures forming the oral cavity and discuss the involvement of the purinergic signal in pathological conditions of the oral apparatus as well as the first experimental findings on the use of drugs interfering with the purine system to improve health conditions during some oral diseases. The review is comprehensive and timely. There are some suggestions to improve the manuscript.

In page 11 line 413 the authors stated:

The most frequent cause of inflammatory alterations in the oral cavity is due to bacterial/viral infections, which mostly concern periodontal tissues. Thus, pathogenic microorganisms, mainly in chronic periodontitis, induce a host inflammatory response mediated by immune cells, which are the principal responsible for ATP release. This ATP, interacting with P2X7R on local fibroblasts or osteoblasts, activates the expression of receptor activator of nuclear factor k-B ligand (RANKL), inducing differentiation and activity of osteoclasts to modify the alveolar bone crest morphology together with cytokines like IL-1ß and tumor necrosis factors α (TNFα) that are known to activate osteoclastic bone resorption.

I suggest the authors enrich their discussion by incorporating data about the involvement of canonical and non-canonical inflammasomes in the purinergic signaling participation in oral diseases.  As suggestion they can read:

-       De Andrade KQ et. al. Curr Res Microb Sci. 2021 Feb 23;2:100023.;

-       Almeida-da-Silva C et. al. Immunobiology. 2019 Jan;224(1):50-59.

- Ramos-Junior E et. al.  J Dent Res. 2015 Sep;94(9):1233-42.

Author Response

We thank this reviewer very much for his/her suggestion, which we have tried to follow as best we can. Consequently, the sentences indicated have been modified (see page 10, lines 400-409 and 419-428), as well as the relevant part of the Table 2. In addition, four new references have been added.

We hope that we have fulfilled the reviewer's request.